# Advection and non-climate impacts on the South Pole Ice Core

Tyler J. Fudge[1], David A. Lilien[1,2], Michelle Koutnik[1], Howard Conway[1], C. Max Stevens[1],
Edwin D. Waddington[1], Eric J. Steig[1], Andrew J. Schauer[1], Nicholas Holschuh[3,1]
[1] Earth and Space Sciences; University of Washington, Seattle, WA 98195
[2] Physics of Ice, Climate, and Earth; Niels Bohr Institute, Copenhagen, Denmark
[3] Department of Geology; Amherst College, Amherst, MA 01002
Email correspondence: tjfudge@uw.edu

**Abstract**

The South Pole Ice Core (SPICEcore), which spans the past 54,300 years, was drilled far from an ice divide such that ice recovered at depth originated upstream of the core site. If the climate is different upstream, the climate history recovered from the core will be a combination of the upstream conditions advected to the core site and temporal changes. Here, we evaluate the impact of ice advection on two fundamental records from SPICEcore: accumulation rate and water isotopes. We determined past locations of ice deposition based on GPS measurements of the modern velocity field spanning 100 km upstream, where ice of ~20 ka age would likely have originated. Beyond 100 km, there are no velocity measurements, but ice likely originates from Titan Dome, an additional 90 km distant. Shallow radar measurements extending 100 km upstream from the core site reveal large (~20%) variations in accumulation but no significant trend. Water isotope ratios, measured at 12.5 km intervals for the first 100 km of the flowline, show a decrease with elevation of -0.008‰ m$^{-1}$ for $\delta^{18}$O. Advection adds approximately 1‰ for $\delta^{18}$O to the LGM-to-modern change. We also use an existing ensemble of continental ice-sheet model runs to assess the ice sheet elevation change through time. The magnitude of elevation change is likely small and the sign uncertain. Assuming a lapse rate of 10°C per km of elevation, the inference of LGM-to-modern temperature change is ~1.4°C smaller than if the flow from upstream is not considered.

## 1 Introduction

Ice cores provide unique and detailed records of past climate (e.g. Alley et al., 1993; Petit et al., 1999; NorthGRIP, 2004; Marcott et al., 2014). Such records are most useful if they represent the change in climate at a fixed geographic location and elevation. Two important non-climatic influences on ice-core records are changes in ice-sheet elevation (Vinther et al., 2009; Steig et al., 2001; Stenni et al., 2011; Parennin et al., 2007; Cuffey and Clow, 1997) and changes in the location of ice origin due to flow (Whillans et al., 1984; Huybrechts et al., 2007; NEEM, 2013; Steig et al., 2013; Koutnik et al., 2016). Many ice cores are drilled near an ice divide to minimize both of these effects: ice thickness varies less in the interior than on the margins (Cuffey and Paterson, 2010) and there is little lateral ice flow near a divide. The change in ice thickness can be evaluated with ice-flow models (Parrenin et al., 2007; Golledge et al., 2014; Briggs et al., 2014; Pollard et al, 2016) or measurements from the ice core itself (Martinerie et al, 1994; Steig et al., 2001; Vinther et al., 2009; Waddington et al., 2005; Price et al., 2007). The magnitude and sign of the elevation change in ice-sheet models varies depending on the specified boundary conditions and model parameters, which have a large uncertainty (DeConto and Pollard, 2016; Kingslake et al., 2018). We assess the ice-sheet elevation change near South Pole in this paper using the 625-member ensemble of the Penn State ice-sheet model (Pollard et al., 2016). We also focus on the impact of ice flow on the South Pole Ice Core (SPICEcore). We will use the term "advection impact" to refer to variations in the ice-core histories that are due to variations in the deposition location and paleo-elevation for different parcels of ice in the South Pole core, as opposed to temporal change in the climate at the ice-core site.

Ice cores are often drilled far enough from divides that lateral advection is important because of site characteristics (NorthGRIP, 2004; EDML, EPICA 2006; WAIS Divide, Morse et al., 2005; NEEM, 2013), logistical considerations (Camp Century, Gow et al., 1968; Dye-3, Dansgaard et al., 1969; Byrd, Hammer et al., 1980; Vostok, Lorius et al., 1985), or concern about divide migration over the drill site (Waddington et al., 2001). The importance of advection on ice-core records depends on both the velocity of the ice and the gradient in the constituent or property of interest. For well-mixed atmospheric gases, such as carbon dioxide and methane, there is no direct impact on the histories. The affected histories are primarily those recovered from the ice phase: accumulation rate, water isotopes, surface temperature, and aerosols. Of the cores that have been drilled off of ice divides, the horizontal velocities range from less than 1 m a$^{-1}$ (EDML) to 12 m a$^{-1}$ (Dye 3) and all require correction to obtain the climate history for a fixed geographic location (Whillans et al., 1984; Steig et al., 2001; Huybrechts et al., 2007; Vinther et al., 2009; NEEM, 2013; Steig et al., 2013; Koutnik et al., 2016).

The 1750 m long SPICEcore was obtained at the South Pole between 2014 and 2016. SPICEcore was sited, in part, to take logistical advantage of South Pole station where the surface velocity is 10 m a$^{-1}$ in the direction of 40°W (Hamilton, 2004; Casey et al., 2014). Lilien et al. (2018) inferred the flowline out to 100 km upstream and concluded that Titan Dome is the likely source region for ice reaching the SPICEcore site. Previous measurements of water isotope values upstream of South Pole are primarily from surface snow samples, which do not provide reliable time-averaged values (Masson-Delmotte et al., 2008; Dixon et al., 2013). A shallow ice core near Titan Dome (US-ITASE 07-4) provides a single estimate of accumulation (0.074 m ice equivalent a$^{-1}$; Dan Dixon, personal communication). Here, we assess the advection impact (i.e.,

non-climate impact) on the accumulation-rate, water-isotope, and surface-temperature histories
of SPICEcore using new measurements in the upstream catchment.


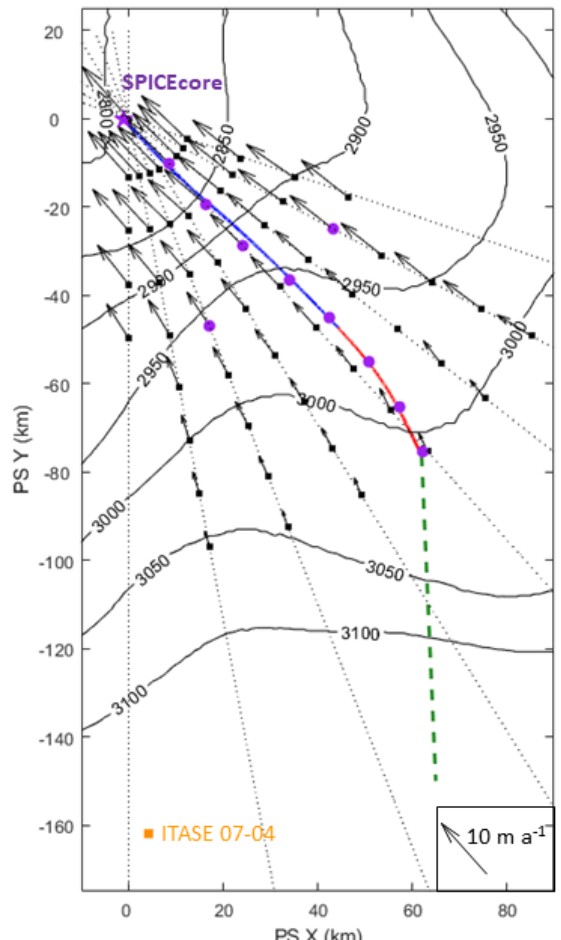

Figure 1: Map of the area upstream of
the South Pole. SPICEcore location is
purple star. 10 m core locations are
purple circles. Stake locations (black
squares) were surveyed with GPS in
multiple years to measure velocity
vectors. Flowline was inferred from the
velocity measurements for past 10.1 ka
(blue, from Lilien et al., 2018) and 10.1
ka to ~25 ka (red). Unconstrained
flowline for ~25 ka to 55 ka is dashed
green. Surface topography contours are
from BedMap2 (Fretwell et al., 2013).
ITASE 07-04 core at Titan Dome is
orange square. Note that Titan Dome is
a broad ridge and the geometry is not
well defined in BedMap2; the elevation
does not match the 3090 m measured by
Dixon et al. (2013).
**2 Methods**
To assess the impact of advection on the SPICEcore climate histories, we measured ice velocity,
accumulation rates, water isotopes, and firn temperatures in the upstream catchment. The surface
ice-flow velocities, inferred flowline, and spatial pattern of accumulation were described by
Lilien et al. (2018; http://www.usap-dc.org/view/dataset/601100) and we provide only a brief
review below.

**2.1 Surface Ice-flow Velocity and Flowline Determination**
Determining the ice-flow velocity near South Pole is more difficult than many other
locations in Antarctica; there is little satellite coverage due to the geometry of satellite orbits
resulting in a data "pole hole." Rignot et al. (2011) used synthetic aperture radar to compute the
surface velocity, but utilized a substantially tilted satellite view, resulting in velocity
measurements that are not sufficiently precise to define the flowline. To obtain improved
velocity measurements in the region, we performed repeat surveys of stakes with GPS during
four consecutive field seasons. We installed 56 stakes at 12.5 km intervals along lines of
longitude from 110°E to 180°E at 10° intervals (Lilien et al., 2018). The 110° and 180° lines
were measured only to 50 km from South Pole; the others were measured to 100 km (Figure 1).
The measured velocities range from 3 to 10 m a$^{-1}$, with errors of ±0.02 to 0.25 m a$^{-1}$ in each
horizontal direction.
**2.2 Accumulation Rate**
The accumulation rate along the flowline is derived from radar layers imaged from
approximately 20 m to 100 m depth with a 200 MHz radar (details can be found in Lilien et al.,
2018). The depth of a radar layer is converted to an accumulation rate using the density profile
and depth-age relationship of a core extracted by us on the flowline 50 km upstream from
SPICEcore. The firn depth-density profile is assumed to be unchanging along the flowline. The
firn density affects the derived accumulation rate history both through the inferred depth of the
layer due to the radar-wave propagation speed and through the conversion to ice-equivalent
thickness. These two uncertainties oppose each other but do not necessarily cancel. Using four
additional density profiles near South Pole, Lilien et al. (2018; Figure S4) found the spread in
accumulation has a standard deviation of 2.3% for a layer at ~20 m depth. Deeper layers have a
smaller spread because the density is most variable near the surface. All accumulation rates are
given in m a$^{-1}$ of ice equivalent.
**2.3 Water Isotopes**
Water isotopes ratios of $\delta^{18}$O and $\delta$D were measured in cores of approximately 10 m depth at
12.5 km spacing along the flowline, as well as at two sites 15 km perpendicular to the flowline
50 km upstream of SPICEcore, for a total of 10 firn cores. We also report the deuterium excess,
using the log definition (d$_{ln}$; Markle et al., 2017).  The cores were sampled at 0.5 m intervals in
the field and allowed to melt in plastic bottles. The measurements were performed at the
University of Washington's Isolab with a Picarro L-2120i. The average $\delta^{18}$O and $\delta$D values (vs
Vienna Standard Mean Ocean Water) for each core are presented here. The cores were not dated
and thus the water isotopes cannot be averaged over the same ages; averaging using only the
upper 5 m for each core instead of the full core produced negligible differences. One outlier from
0.5-1 m depth at site 25 km was excluded.
**2.4 10 m Temperatures**
The temperature at approximately 10 m depth was measured in each borehole left by the
shallow-core extraction. We averaged the values measured by four thermistors surrounded by a
copper shield. The thermistors were left in the borehole for different lengths of time ranging
from 28 minutes to 48 hours.
**2.5 Analysis of Continent-scale Ice Sheet Models**
We use a 625-member ensemble of the Penn State ice-flow model (Pollard et al., 2016) to assess
possible ice-sheet changes during the deglacial transition. The model uses a 20-km grid size for
West Antarctica, which includes the South Pole region. The accumulation rate applied at 20 ka is
approximately half of the modern value (Pollard and DeConto, 2012). The ensemble is used to
assess the histories of surface velocity and elevation of the South Pole. The ensemble varies four
different ice-dynamic parameters with five values each. The four parameters affect the basal
sliding coefficient where ice is no longer grounded (CSHELF); ice shelf melt rate (OCFAC);
calving rate factor (CALV); and isostatic rebound (TAUAST). We perform evaluations using
both the full ensemble (n=625) and a subset, including only the parameter values identified with
the advanced statistical techniques (n=32) to best fit geologic constraints (Table 1; Pollard et al.,
2016, Figure 3, right column).

Table 1: Pollard et al. (2016) most likely
parameter values

| Parameter | Abbreviation | Value | Unit |
|---|---|---|---|
| Basal sliding coefficient in modern oceanic areas | CSHELF | -6 and -5 | $10^x$, m a$^{-1}$ Pa$^{-2}$ |
| Bedrock-elevation isostatic relaxation time | TAUAST | 1, 2, 3 and 5 | kyr |
| Calving rate factor | CALV | 1 and 1.3 | non-dimensional |
| Melt-rate coefficient at base of ice shelves | OCFAC | 1 and 3 | non-dimensional |

**3 Results**
**3.1 Gradients in Upstream Climate**
**3.1.1 Accumulation Rate**
The accumulation rate along the 100 km flowline for four different internal layers is shown in
Figure 2. The youngest layer is 151 years before 2017 (~20 m depth) and was used by Lilien et
al. (2018); the 743-year layer is the deepest (~90 m) layer resolved. Although the layers are
relatively young, there can still be a horizontal offset of hundreds of meters to kilometers from
where the layer was deposited on the surface. In Figure 2A, the accumulation rates in the upper
panel are plotted at the position of the radar trace. The impact of horizontal advection can be
observed as the older layers appear shifted to the left (closer to SPICEcore) compared to the
younger layers.
To account for horizontal advection, the position where the accumulation rate is inferred (i.e. the
location of the radar trace) is adjusted. This adjustment is made by multiplying the half-age of
the layer by the surface velocity at the mid-point of its path from deposition to the current trace
location (Figure 2B). The adjustment ranges from 3.7 km at SPICEcore for the 743-year layer to
0.2 km for the 151-year layer at the upstream end. Shifting the distance of the accumulation
records (Figure 2C) better aligns the peaks and troughs among the four layers. It also highlights
that older layers vary less along flow. The depth of a layer reflects the average surface
accumulation rate over the distance traveled. Thus, an older layer is flatter because it averages
the influence of accumulation on vertical velocity over a longer distance (Waddington et al.,
2007). This shows that simply shifting the position of the layers to account for horizontal
advection does not fully recover the spatial variations in accumulation.
A more-complete treatment could solve an inverse problem to infer the surface accumulation rate
along the flow line that best matches the observed layer thicknesses (e.g., Waddington et al.,
2007). We do not address this because here we focus on the advection impact on the SPICEcore
record and not a formal evaluation of the surface accumulation patterns consistent with available
layers. Lilien et al. (2018, supplement) showed that the 151-year layer was sufficiently deep to
record real climate variations, and not noise, but shallow enough to not be significantly affected
by lateral flow.


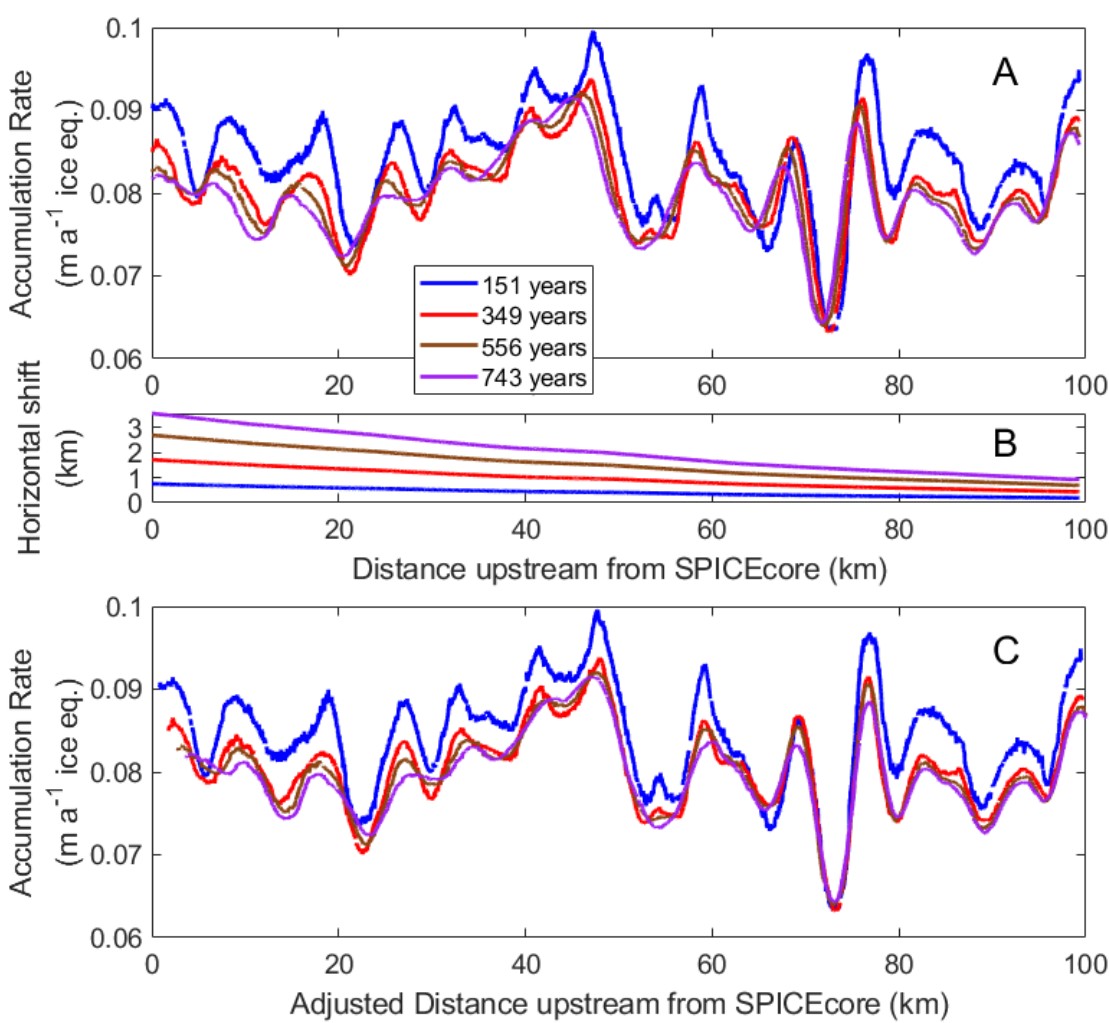

Figure 2: Accumulation rate along flowline. Panel A shows the accumulation rate for four radar
layers, with ages in years before 2017. Panel B shows average horizontal distance traveled. Panel
C shows same inferred accumulation as in Panel A, with the position adjusted to account for the
horizontal distance traveled.

The average accumulation rate of the oldest (743-year-old) layer is 0.080 m a$^{-1}$ and the spatial
linear trend of -4×10$^{-6}$ m a$^{-1}$ km$^{-1}$ is negligible. Shorter-wavelength spatial variations are
approximately ±20% of the average value, much larger than the linear trend. Beyond the 100 km
of mapped flowline, the only accumulation-rate information is from the US-ITASE 07-04 core
near Titan Dome, where an accumulation rate of 0.074 m a$^{-1}$ was inferred (Daniel Dixon,
personal communication, 2013). This is within the range of accumulation rates identified along
the flowline, but slightly smaller than the 0.080 m a$^{-1}$ average along the first 100 km of the
flowline. With only a single point measurement, we cannot resolve whether this accumulation
rate near Titan Dome is representative of a mean value for a wider area.

We also calculate the accumulation rate for the intervals between successive layers (Figure 3),
which allows temporal trends to be more clearly evaluated. The uncertainty in the accumulation
rate is greatest for the 151-year layer because the density measurements are least certain in the
lower-density surface snow, and surface firn conditions are more spatially variable. We calculate
the uncertainty for an interval based on the density profiles of five different firn cores (the core
we drilled at 50 km and four cores from near South Pole; Severinghaus et al., 2001; Christo
Buizert, personal communication). The uncertainty shading shown in Figure 3 is the range
between the maximum and minimum accumulation rates using the five density profiles. The
spatial average of the three older intervals are within uncertainty of each other. The spatial
average of the 0 to 151-year interval is always greater than the older three intervals. Because the
spatial average of the minimum accumulation rate (based on firn density) for 0 to 151-year is
greater than the spatial average of the maximum for the older intervals, we have confidence that
the accumulation rate has increased in the past 151 years. The accumulation increase is 8±4%
compared to the previous 592 years (151 to 743 years before 2017). Previous ice-core estimates
of accumulation at South Pole suggested an increase in the past 150 years (e.g. Ferris et al.,
2011), but an increase could not be identified with confidence because variations among cores
were dominated by spatial, not temporal, effects (van der Veen et al., 1999). Our measurements
average over a 100 km distance allowing the temporal change to be identified.

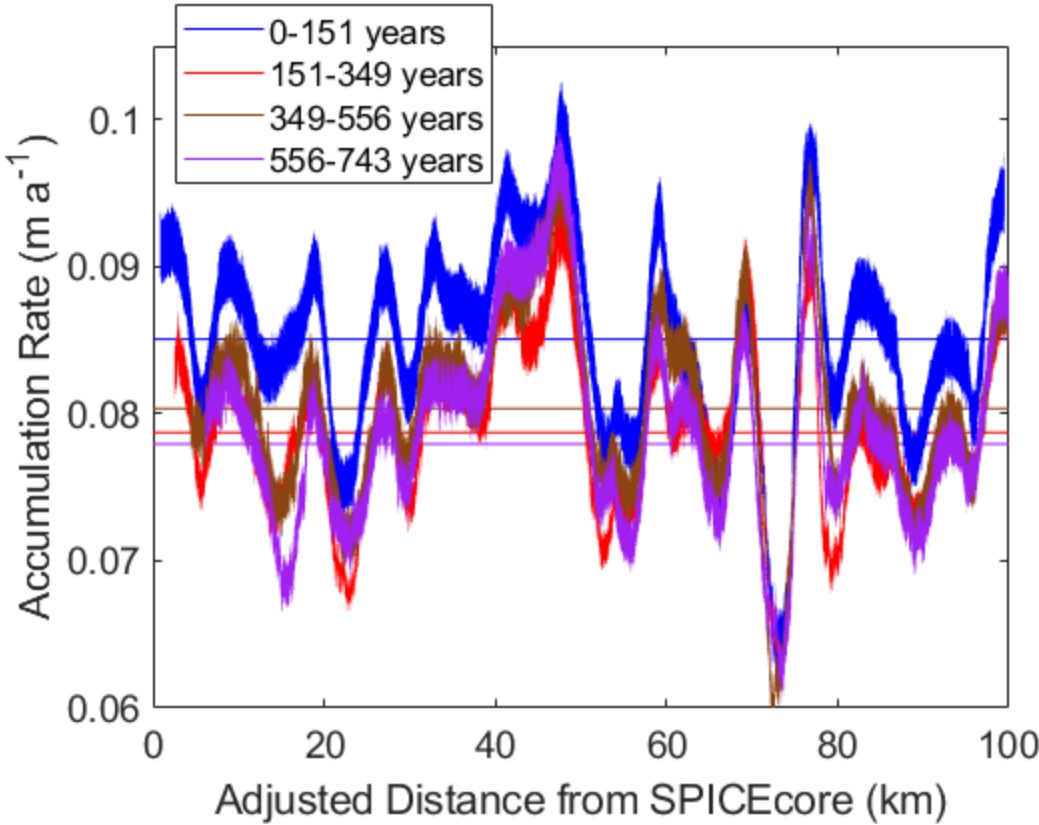

Figure 3: Temporal average accumulation rate for ages between radar layers. Shading indicates
uncertainty based on five firn-density profiles. Distance from SPICEcore has been adjusted as in
Figure 2 and described in main text. Horizontal lines indicate spatial average of the accumulation
rate using the density profile measured on the firn core at 50 km.

| Table 2: Accumulation Increase in past 151 years relative to previous periods | | | |
|---|---|---|---|
| Interval | Mean | Minimum | Maximum |
| 151-349 | 8% | 4% | 12% |
| 349-556 | 6% | 1% | 11% |
| 556-743 | 9% | 3% | 13% |
| **151-743** | **8%** | **4%** | **12%** |

Mean increase uses density profile from the core at 50km for all layers

Minimum (maximum) increase uses density profile which yields the minimum (maximum) accumulation rate for the 0-151 interval and the density profile which yields the maximum (minimum) for the older layers.

### 3.1.2 Water Isotopes

Measurements of water isotopes require the collection of ice samples and thus have less spatial resolution than the radar-derived accumulation-rate measurements. There is considerable scatter (Figure 4) in the 0.5 m resolution samples, which have durations of a few years (i.e. 2-4 years) per sample; the differences among 0.5 m samples are likely driven by interannual variations. Using the mean values, a decrease with distance from South Pole is observed in both $\delta^{18}O$ and $\delta D$. The $d_{ln}$ values show no significant trend upstream.

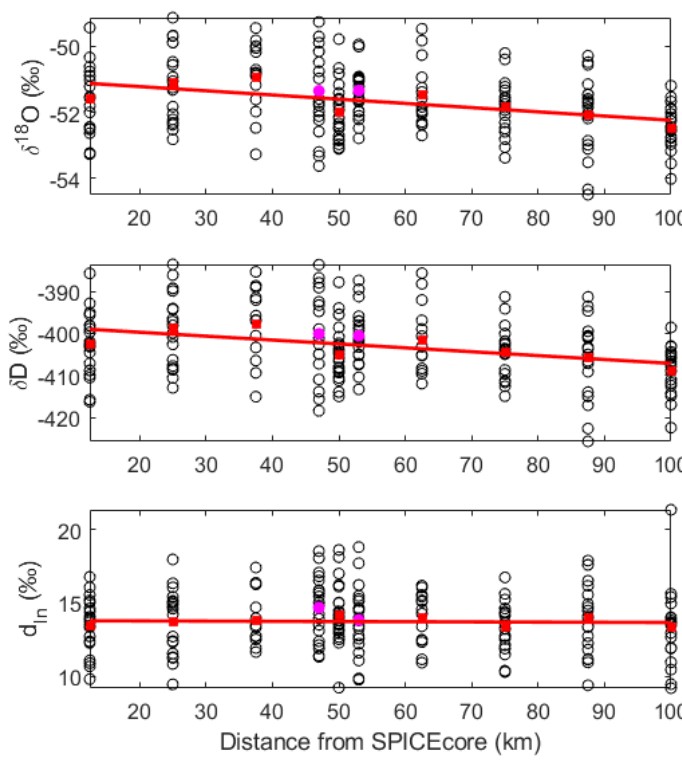

Figure 4: Water-isotope values (black circles) and averages (red squares) for shallow cores along the flowline upstream of South Pole. Cores at 50 km upstream on 120°E and 160°E are plotted at 47 km and 53 km (magenta circles). Linear slope (thick red line) is from the average values along the flowline only.

The $\delta^{18}$O and $\delta$D values plotted by elevation are shown in Figure 5. Linear fits to $\delta^{18}$O and $\delta$D
yield slopes of -0.0080 ± 0.0055 ‰ m$^{-1}$ and -0.0579 ± 0.04 ‰ m$^{-1}$ respectively (95% confidence
levels). Our value for $\delta^{18}$O is in between the slope of -0.009 ‰ m$^{-1}$ from the Masson-Delmotte et
al. (2008) database and the slope of -0.007 ‰ m$^{-1}$ found in their multiple linear regression
analysis which includes latitude and distance from the coast. Including the average $\delta^{18}$O value
from the upper 1.2 m of the US-ITASE 07-04 firn core at Titan Dome (-53.15‰) in the linear
regression changes the slope to -0.0073‰ m$^{-1}$, which is in good agreement with the mean slope.
Because the Titan Dome value is an average of the upper 1.2 m and not directly comparable in
time to our 10 m average measurements, we use the mean slope of 0.008 ‰/m$^{-1}$ from the 10 m
cores for the advection correction described in the subsequent section.

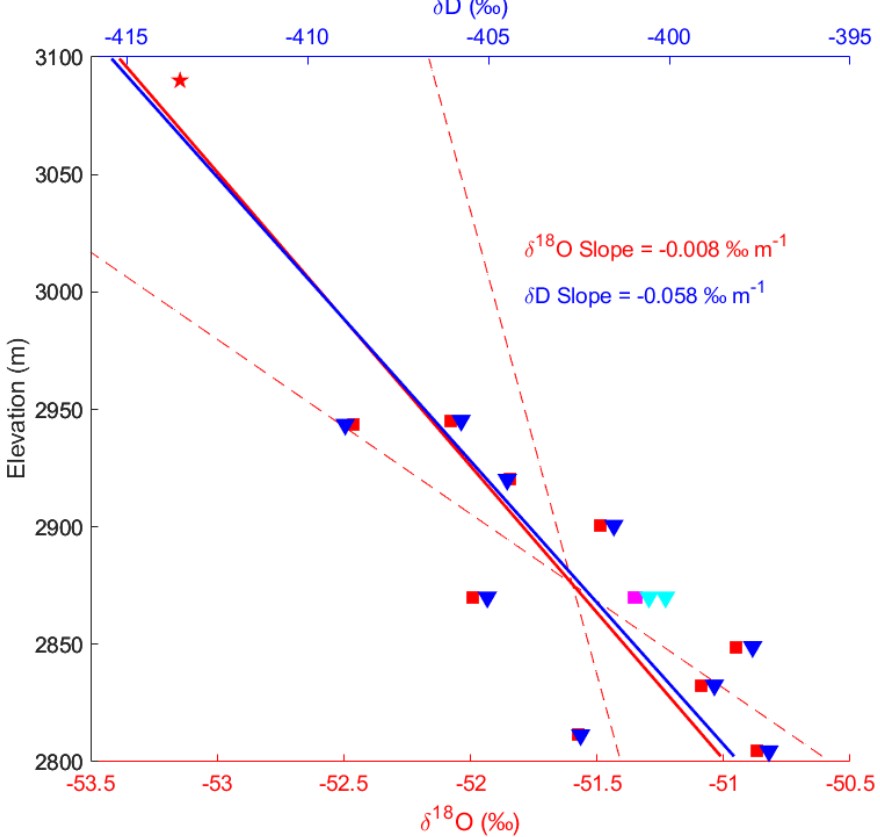

Figure 5: Average $\delta^{18}$O (red squares) and $\delta$D (blue triangles) values from the 10 m cores along
the flow line and SPICEcore. Average $\delta^{18}$O and $\delta$D from cores off of the flowline at 50 km
upstream (pink squares and cyan triangles). $\delta^{18}$O of US-ITASE 07-04 core at Titan Dome (red
star). Linear fit of 10 m cores along the flow line for $\delta^{18}$O (red thick line) and $\delta$D (blue thick
line) do not include Titan Dome or cores from off the flowline. 95% confidence intervals of the
$\delta^{18}$O fit (red dashed lines) are shown. Confidence intervals of $\delta$D overplot those of $\delta^{18}$O and are
not shown.
**3.1.3 Surface Temperature Gradient**
The ~10 m temperatures are shown in Figure 6. Unfortunately, time constraints in the field
forced differences in the measurement procedure between sites, preventing a determination of
the gradient in mean annual temperature. Measurements that equilibrated for less than 1.5 hours
yielded warmer temperatures than those left in boreholes for longer times, and we consider those
shorter measurements less reliable. Measurements that were made after leaving the thermistors in
the boreholes for longer than six hours are consistent with a dry adiabatic lapse rate of 10°C km⁻
¹, but we cannot reject a wide range of other values for the lapse rate.

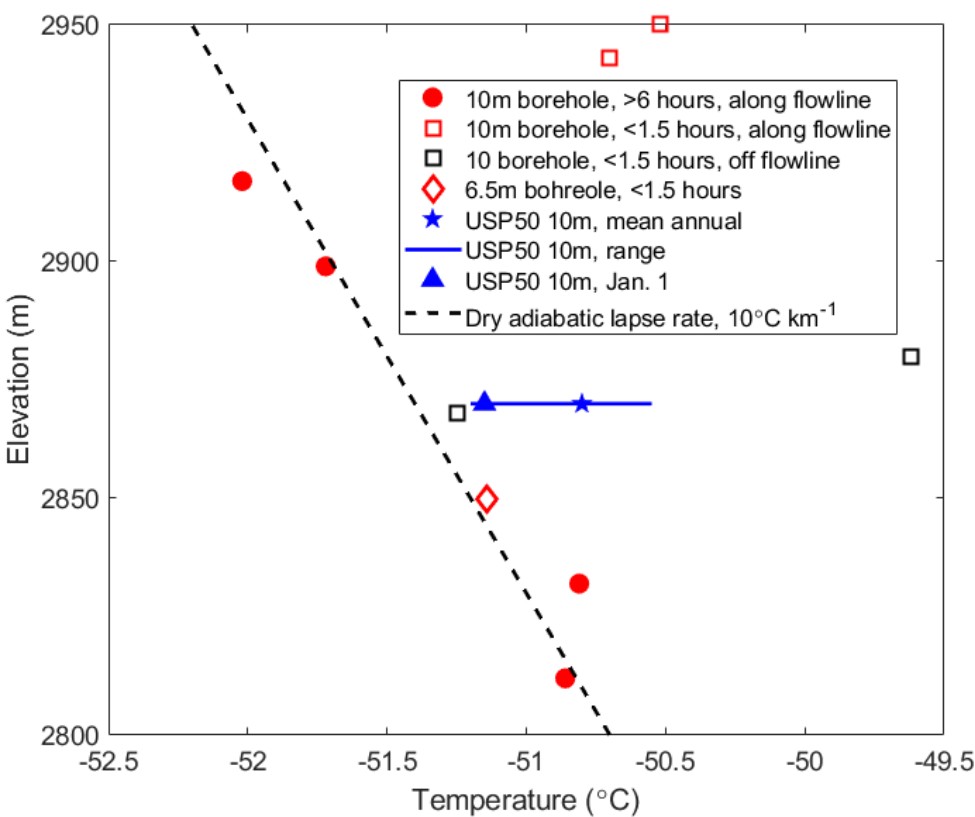

Figure 6: Temperature measurements. Filled symbols equilibrated for more than 6 hours; open
symbols equilibrated for less than 1.5 hours. Red symbols are along the flow line; black symbols
are off the flowline. Diamond is a measurement at 6.5 m depth, which is likely ~0.7°C colder
due to the winter cold wave than if measured at 10 m depth. Blue symbols are from a single
thermistor installed at 10 m depth in a back-filled borehole with measurements recorded for more
than 1 year; star is mean annual temperature, triangle is initial temperature after equilibration and
horizontal line is the range of temperature recorded. Black dashed line shows a lapse rate of
10°C km⁻¹.
**3.2 Determination of Flowline Position and Age**
We divide the reconstruction of the flowline into three segments based on the data available for
different distances upstream from SPICEcore:
1) 0 to 65 km (0 to 10.1 ka) which has been constrained by Lilien et al. (2018)
2) 65 to 100 km (10.1 to ~25 ka) where we have velocity measurements
3) beyond 100 km (older than ~25 ka) where only limited data from other sources exist
The uncertainty associated with the reconstruction increases for each segment because of the
data available as well as possible changes to the ice-sheet configuration at earlier times. For
segment 1, the uncertainty is low because correlation of the SPICEcore layer thicknesses and
upstream accumulation pattern provides a unique and tight constraint (Lilien et al., 2018). For
segments 2 and 3, we have no inferences of past ice-sheet velocity. The variation in horizontal
velocity with depth does not need to be considered because we are only interested in tracking
particles to 1750 m depth in SPICEcore where the modeled horizontal velocity is at least 99% of
the surface velocity. The challenge of determining the flowline position with age is then of
estimating the past surface velocity. The modeled surface velocities near South Pole in the ice-
sheet ensemble (Pollard et al., 2016) are slower than observed (mean of 2 m a$^{-1}$ for the models
runs compared to the measured 8 m a$^{-1}$ at ~20 km from SPICEcore) and thus cannot be used
directly. Instead, we use the relative change in speed between 20 ka and 10 ka to inform our
choice of speed change for this time period. The full ensemble (Figure 7) shows a large fraction
of model runs with faster velocities at 20 ka compared to 10 ka with a mean slowdown of 10%
from 20 ka to 10 ka. The speed changes in the limited ensemble are bimodal: one group shows
speeds at 20 ka between 50% and 90% of the speed at 10 ka. The other group shows between no
change and 10% faster speeds at 20 ka compared to 10 ka. The first group is closer to the speed
that might be expected if the speed was primarily determined by the accumulation rate through a
balance velocity; the second group indicates that dynamic changes are able to counteract the
influence of lower accumulation rates at 20 ka. We thus determine the speeds for ages older than
10.1 ka in two ways: no change in speed and speed changes that scale with an approximate
accumulation history.

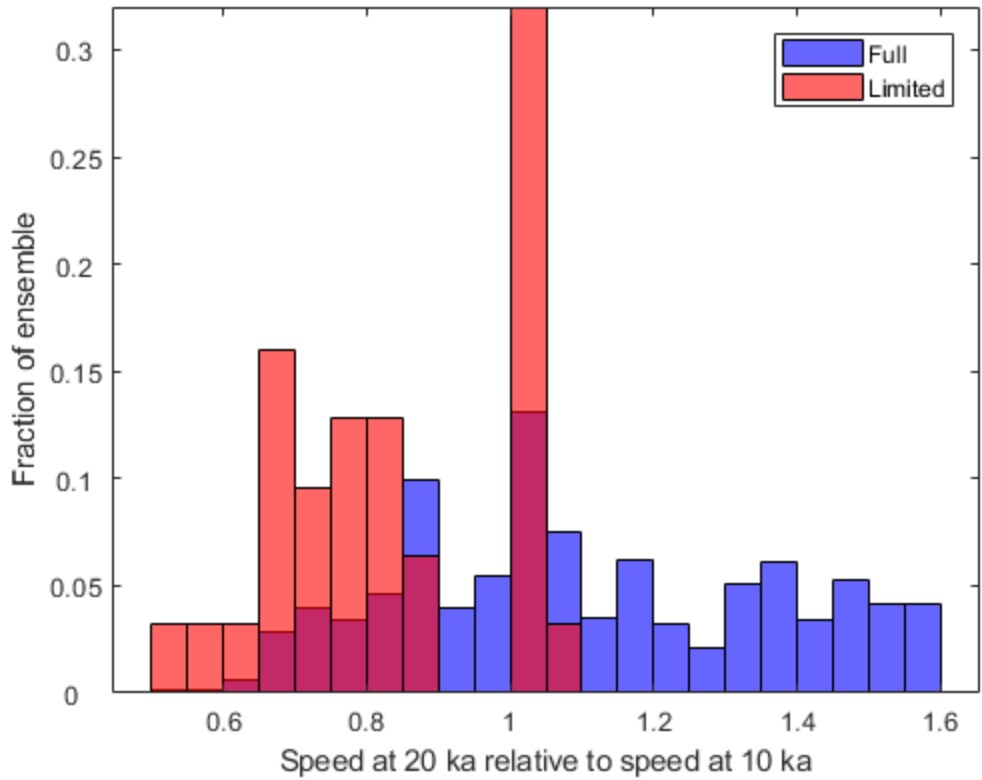

Figure 7: Histograms of modeled speed changes between 10 ka and 20 ka near South Pole for the
full and limited ensembles (see 2.5 for full description; Pollard et al. 2016)
**3.2.1 Segment 1: 0 km to 65 km (0 to 10.1 ka)**
The first segment uses the inferred flowline of Lilien et al. (2018). They used a novel method of
correlating the SPICEcore layer thicknesses with the geophysically-determined accumulation
pattern upstream and found that with a 15% increase in speed from 10.1 ka to today, the
upstream pattern of accumulation explained approximately three-quarters of the variance in the
SPICEcore accumulation history. Of particular importance to this study, their work tightly
constrains the location where the ice in the core was deposited on the surface of the ice sheet.
This has not been possible at previous ice-core sites (e.g. WAIS Divide, EDML, NEEM) where
ice-flow models provided the only estimates of past velocity.
The measured velocity field was used to determine the modern flowline. We use the flowline
position and age from the preferred scenario of a 15% Holocene speed up of Lilien et al. (2018).
The position and age were found by starting at the SPICEcore drill site and recursively stepping
upstream in one-year intervals in the direction opposite the velocity vectors to obtain annual
positions along the flowline. The velocity direction was fixed in time while the magnitude was
linearly decreased to 15% slower velocities at 10.1 ka. The 10.1 ka ice originated 65 km
upstream along the flowline.
**3.2.2 Segment 2: 65 to 100 km (10.1 to ~25 ka)**
For ice older than 10.1 ka, the spatial variations in the accumulation rate cannot be clearly
correlated with the layer thickness variations in SPICEcore. This is likely because: 1) uncertainty
in the flowline position increases with distance (age); 2) the relative uncertainty in the surface
velocity increases as the velocity decreases with distance upstream; 3) the surface-velocity
measurement stakes are farther apart; and 4) the temporal variations in accumulation are likely
larger during the isotopic maximum at ~11 ka and the glacial-interglacial transition (Veres et al.,
2013; Fudge et al., 2016). This segment of the flowline spans from 65 km to the limit of the
surface velocity measurements at 100 km from the SPICEcore drill site. Without the constraints
of the correlation analysis, both the flow direction and past ice-flow velocity are much less
certain. Continent-scale ice-sheet models have difficulty reproducing the details of ice-flow in
the region and are sensitive to boundary forcing assumptions.
We use two different assumptions about the past ice speed to estimate the flowline position with
age before 10.1 ka. For both methods, we start with the inferred speed at 10.1 ka from Lilien et
al. (2018; i.e. 15% slower than measured today) and keep the ice-flow direction fixed in time.
The first reconstruction assumes that the speed has been constant in time prior to 10.1 ka.  The
second reconstruction scales the speed to an estimate of the past accumulation rate, essentially
assuming that the speed is controlled by the ice flux necessary to keep the ice sheet in balance.
The speed history used is shown in Figure 8. Winski et al. (2019) only reported the SPICEcore
accumulation history for the Holocene (younger than 11.7 ka) because the cumulative thinning
layers have experienced becomes increasingly uncertain with depth. Since we are only seeking a
plausible estimate of past speed, the increased uncertainty of the thinning function is not a major
concern for this work. We obtain an accumulation history for the past 54 ka by dividing the layer
thicknesses of the SP19 timescale (Winski et al. 2019) by a thinning function computed with a
Dansgaard-Johnson (1969) model of vertical strain with a kink height of 0.2 and low pass filtered
at 5 ka. Scaling the ice-flow speed to the accumulation rate results in speeds at the LGM of only
40% of modern; thus, ages at the end of the measured flowline, at 100 km from SPICEcore, are 7
ka older (28 ka) than with the assumption of a constant speed (21 ka).

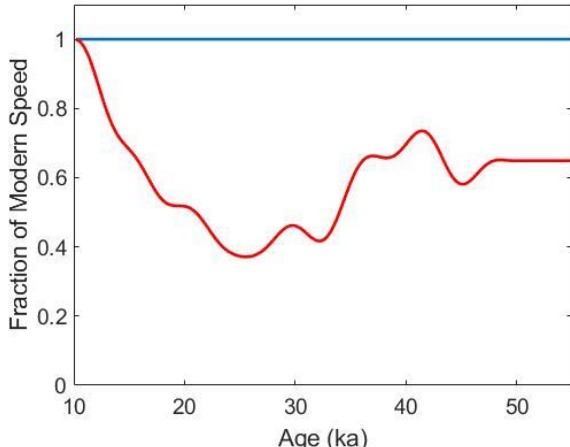

Figure 8: Fraction of modern speed used to reconstruction flowline position and age for the
constant speed scenario (blue) and scaled to accumulation history (red).
**3.2.3 Segment 3: beyond 100 km (older than ~25 ka)**
For ice that originated beyond 100 km from SPICEcore, no reliable surface-velocity
measurements exist to help define where the ice originated. We examined the utility of the
surface topography of BedMap2 (Fretwell et al., 2013) in defining the flow direction by tracking
particles along the steepest descent. We computed two flowlines, one going upstream from
SPICEcore and the other going downstream from the 10 ka location. They do not agree with
each other or with the measured flowline, which is not surprising given the limited data in
BedMap2 and the convergent flow. Thus, we do not expect the surface topography to be useful
in defining the x and y components of the flowline beyond 100 km and we assume that the ice
has flowed in a straight line from an ice divide (Figure 1). The position of the ice divide is not
well defined and we assume it is an additional 90 km distant. We also assume that the speed
decreases linearly from its value at 100 km to zero at the divide, equivalent to assuming a
balance velocity in an ice sheet with uniform ice thickness and accumulation rate and no
convergence or divergence, because we have little information of the bedrock topography
upstream. We then apply the same two assumptions for the flow speed used for the second
segment: either constant speed or varying based on the accumulation history. These assumptions
suggest the oldest SPICEcore ice (54.3 ka) originated a total of 135 km to 155 km upstream from
SPICEcore.
**3.3 Advection Impact**
The advection impact on the SPICEcore accumulation-rate and water-isotope histories are quite
different from each other. The accumulation rate is sampled with high frequency but shows no
long-term trend with distance and elevation. The water isotopes, on the other hand, are sampled
infrequently but show a linear trend with distance and elevation. We discuss the advection
impact for the two separately.
**3.3.1 Accumulation Rate**
The lack of a linear trend in the accumulation rate along the flowline indicates that no trend
should be removed from the SPICEcore accumulation history. However, the variation in
accumulation upstream has a major impact on the SPICEcore history. Lilien et al. (2018) were
able to isolate the influence of km-scale upstream variability for the past 10 ka, which explains a
majority of the variance in the SPICEcore accumulation history. Thus, little of the variability in
the accumulation history for the past 10 ka is due to climate. While the residual variance of the
SPICEcore accumulation history (the accumulation history after removing the advection impact)
might reflect temporal changes in climate, the residual variance is also affected by multiple
sources of uncertainty such as the assumptions of a constant spatial pattern of accumulation, a
fixed flowline, a linear speed up, and a spatially homogeneous firn-density profile. These
uncertainties are sufficiently large and difficult to quantify that we do not interpret the residual as
a temporal history of accumulation.
Beyond 10 ka, it is important to understand the potential influences of spatial variations in
accumulation in order to avoid erroneous conclusions about temporal variations in the
accumulation rate over the past 54 ka. Since there is no overall trend, we are primarily interested
in how the spatial variability could be imprinted in the ice-core history. Spectral analysis of the
spatial pattern of accumulation shows that there is significant power at a wavelength of 5 to 10
km. The temporal imprint of the spatial variations on ice-core derived accumulation rates is then
determined by the ice-flow velocity, which is 4 m a$^{-1}$ for ice of 10 ka age and decreases to 1 m a$^{-1}$
for ice of 54 ka age. The timescales affected in the accumulation history are ~1 to 6 ka during the
deglacial transition (10-20 ka) and get longer, reaching 10 ka, for the glacial SPICEcore ice. The
advection impact on the deglacial transition may affect the specific timing of accumulation-rate
change, but not the overall temporal trend. For older ages, the advection impact has a similar
timescale to millennial-scale climate variations. We thus expect that the advection impact will
decrease the coherence between the accumulation-rate history and the temperature history
inferred from water isotopes.
**3.3.2 Water Isotopes**
The water isotopes are not sampled at a high enough spatial resolution to perform an analysis of
millennial-scale variations as was done for the accumulation rate; however, the $\delta^{18}O$ and $\delta D$ both
show linear trends with elevation and distance. Because $\delta^{18}O$ and $\delta D$ are similar, we will discuss
only the advection correction for $\delta^{18}O$ in this section (both are provided in the supplemental
spreadsheet). A correction for advection becomes important, particularly for questions such as
the magnitude of the glacial-interglacial temperature change. We use a linear fit to elevation data
as the base for the advection correction (Figure 9). The linear fit is continued beyond 100 km at
the same slope, reaching an elevation similar to the US-ITASE 07-04 core at 190 km upstream of
SPICEcore. We use the linear fit to avoid meter-scale elevation variability being added through
the advection correction.
We use the two inferences of the origin positions of ice in SPICEcore described in section 3.2 to
find the elevation change through time due to advection. We convert this into an advection
impact for $\delta^{18}O$ based on the linear $\delta^{18}O$-elevation fit (Section 3.1.2; Figure 5), which we assume
is constant in time. The two scenarios provide an estimate of the range of plausible advection
impacts. While we do not have enough information to define a formal uncertainty on the
advection impact, the difference between the two scenarios provides a qualitative uncertainty
estimate for the effect of past speed changes. We use the average of these two scenarios as our
best estimate of the advection impact and report all three in the supplementary table.
SPICEcore ice of 20 ka age is approximately 1.1‰ more enriched than if it had fallen at South
Pole instead of at ~95 km upstream and at ~135 m higher elevation. The uncertainty of this
advection impact due to the temporal surface velocity assumption is approximately ±0.1‰;
however, there is additional uncertainty due to the slope of the elevation-water isotope fit.
Because the elevation change is linear with distance, the curvature of the advection impact is
determined by the change in ice velocity and the advection impact increases the most rapidly at
the youngest ages. The difference over the Holocene (past 11.7 ka) is 0.85‰ while the additional
difference to the LGM (20 ka) is only 0.25‰. The advection impact for the oldest ice is only
about 0.01‰ per ka and is nearly the same for both velocity assumptions after 35 ka; this is
because the ice in the constant-speed scenario has moved closer to the divide where the speed is
lower and thus is similar to the lower speed in the accumulation-scaled scenario.

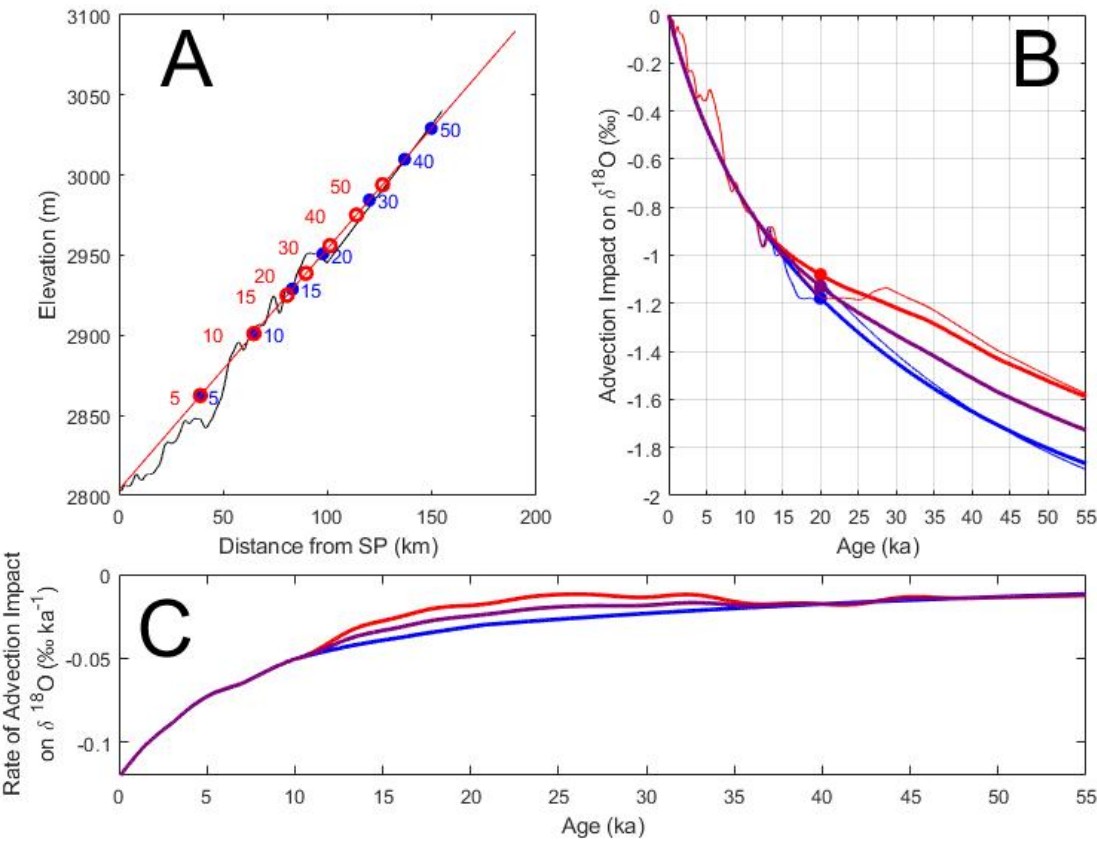

Figure 9: Advection Impact for $\delta^{18}$O. A): Elevation profile (black) and linear fit (red) used in
advection correction. Elevations at 5 ka intervals for the constant velocity assumption (blue dots)
and scale to accumulation history (red circles). B): Advection correction using elevations in left
panel. Blue is constant velocity. Red is scaled to accumulation history. Thick lines use linear
elevation change; thin lines use measured elevation along flowline. The average of the two
assumptions is shown in purple. A negative value indicates the ice recovered in the core fell at a
location where the water isotopes are more depleted than South Pole in the current climate. C)
The rate of the advection impact for the three curves in the upper right panel.
**3.4 Ice Sheet Elevation Change**
The in situ measurements performed in this study provide little in the way of constraints for past
ice thickness change. Lilien et al. (2018) noted that the inferred 15% Holocene speed up could be
caused by either a modest thickening of ~100 m or a steeping of a few percent. However, the
analysis cannot be used for older ages with larger climate changes and potentially more elevation
change. Therefore, we assess the range of plausible elevation change using the output of a 625-
member ensemble of a full ice-sheet model (Pollard et al., 2016) as well as a limited ensemble
(32 members) of the most likely parameter combinations (see section 2.5). We calculate the
mean, median, and standard deviation of the elevation change relative to modern (Figure 10) for
the full and limited ensembles. We note that every member of the limited ensemble has ice
thickness changes of less than 100 m in past 10 ka.
The full ensemble suggests the ice sheet thickened, and the surface elevation increased, from
15ka to 8 ka, before Holocene thinning reduced the ice sheet elevation back to near 20 ka values.
The median change is roughly half the magnitude of the mean with a peak elevation that occurs
at about 10 ka. The limited ensemble shows limited variance about the full model median, with
less elevation change after 8 ka and a slightly higher elevation at 20 ka. The limited ensemble is
bimodal, with the group of runs with a higher elevation at 10 ka corresponding to the basal
sliding coefficient of ungrounded areas parameter (CSHELF) equal to -6 and the group of runs
with lower elevations at 10 ka from runs with CSHELF equal to -5. The maximum elevation
change of the limited ensemble mean is +26 m at 10 ka. The mean elevation is +16 m at 20 ka. In
all cases, one standard deviation encompasses both higher and lower elevations for all past ages
to 20 ka. Therefore, we do not provide an explicit correction for past ice sheet elevation. An
elevation change of 26 m corresponds to a 0.2 ‰ impact for $\delta^{18}O$ using the measured, modern,
spatial slope of 0.008 ‰ m$^{-1}$. This is roughly one quarter of the advection correction at 10 ka.
Thus, uncertainty from possible ice sheet elevation change should be considered in any
interpretation of the water isotope record, but existing ice-sheet models cannot sufficiently
constrain the elevation history to warrant an explicit correction.

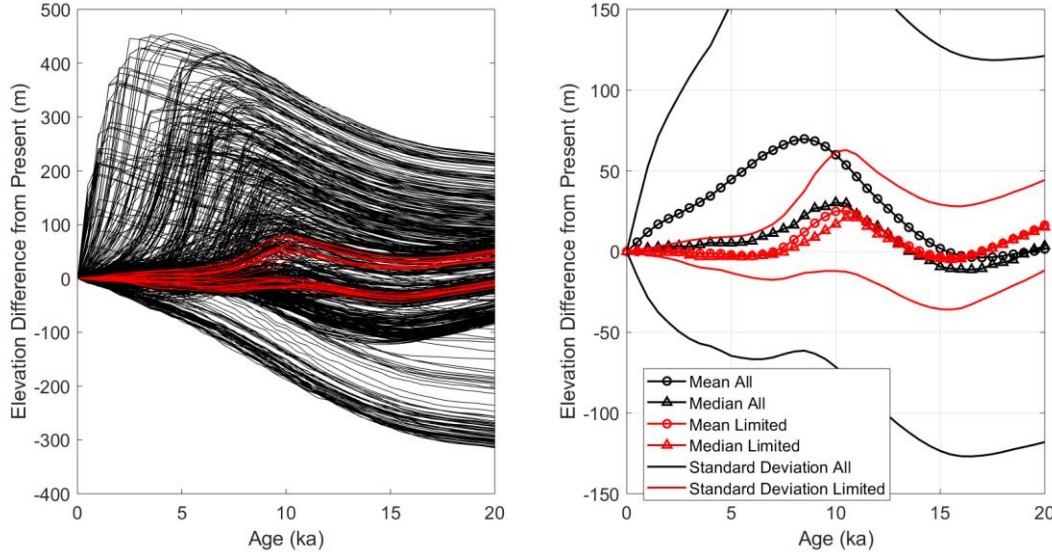

Figure 10: Left Panel: Elevation difference from modern for each model run in the Pollard et al.
2016 ensemble (black, 625 members) and limited ensemble (red, 32 members) of the most likely
parameter combinations. Right panel: mean (circles), median (triangles), and standard deviation
(thin lines) of full ensemble (black) and limited ensemble (red).

**4 Discussion**
Advection has enhanced the glacial-interglacial $\delta^{18}O$ change at SPICEcore by ~1‰ because ice
in the core originated at higher elevations with more depleted isotopic values. The total LGM (20
ka) to modern (past 1 ka) $\delta^{18}O$ change in SPICEcore is approximately 6‰ (Kahle et al., 2018).
Accounting for advection reduces the fixed-location glacial-interglacial change to 5‰.
Advection has the opposite impact at the WAIS Divide ice core (WDC), where advection
increases the glacial-interglacial change by 1‰ to 8‰ (Steig et al., 2013; WAIS Divide Project
Members, 2013). Understanding the advection impact is important for comparing the magnitude
of isotopic change among Antarctic ice cores; WDC has a 1‰ greater LGM-modern change than
SPICEcore in the measured records, but a 3‰ greater change after accounting for advection.
Because SPICEcore and WDC have similar source regions and distillation pathways (e.g.
Sodemann and Stohl, 2009), the difference between the two cores has the potential to yield
insight into the relative elevation change between the West and East Antarctic ice sheets and to
further refine the range of plausible model results presented in Figure 10. A full interpretation of
relative isotopic change between SPICEcore and WDC is beyond the scope of this paper, but
including the impact of advection is critical for future analysis.
The advection impact on the accumulation history is distinct from that for the water isotopes.
There is no linear trend in accumulation in the upstream catchment, and thus no trend to remove
from the SPICEcore accumulation history. High spatial resolution measurements of the modern
upstream accumulation pattern have revealed that the majority of the accumulation variability in
the past 10 ka is caused by advection and not temporal changes (Lilien et al., 2018). While the
upstream pattern and SPICEcore history cannot be correlated for ages older than 10 ka, the
spatial pattern is still expected to impact the accumulation history. The dominant timescales
affected increase from ~1 ka in the Holocene to ~10 ka at 50 ka. These timescales are similar to
that of millennial climate change and thus we expect the spatial variability of accumulation that
is imprinted on the SPICEcore temporal history to decrease the coherence between water isotope
(as a proxy for temperature) and accumulation records. Overall, changes in accumulation of less
than 20% on millennial timescales should not be interpreted as a climate signal.
The different character of the advection impacts for water isotopes and accumulation arise
because there is no coherent relationship between water isotopes and accumulation rate. This
may be because the water isotopes are largely controlled by the condensation temperature
(Jouzel et al., 1997), whereas the accumulation rate is affected by wind redistribution and the
local surface topography (Hamilton, 2004). In fact, the curvature (second derivative) of the
elevation profile along the flowline explains a third of the variance in the modern spatial pattern
of accumulation, similar to areas in Greenland (Miege et al., 2013; Hawley et al., 2014).
The impact of elevation change on the isotopic records is not clear. An ensemble of continental-
scale ice sheet model runs showed minimal mean and median elevation changes in the past. The
standard deviation of the runs always included changes of both signs. Therefore we do not
suggest a correction for ice-sheet elevation change through time but note that there is uncertainty
associated with a possible change that should be considered in subsequent analyses. We also
could not determine the temperature lapse rate from our 10 m borehole temperatures; however,
we can estimate the temperature impact of advection based on a dry adiabatic lapse rate of 10°C
km$^{-1}$, which is consistent with our measurements. The LGM ice fell at ~140 m higher elevation
and likely would be ~1.4°C colder than if it had fallen at the current elevation of South Pole.
**5 Conclusion**
The relatively fast ice speed at South Pole today causes ice at depth in SPICEcore to have
originated at locations up to 155 km away in the direction of Titan Dome and at elevations
upstream of up to 230 m higher, assuming the ice-sheet configuration has not changed
significantly in the past. Elevation change of the ice sheet through time is likely small and of
uncertain sign. Our measurements in the upstream catchment define the flow direction and speed
as well as spatial gradients in the accumulation rate and water isotopes. These measurements
identify the impact of advection on the SPICEcore records. The accumulation rate has no spatial
trend, but shows 20% variations on length scales of 5-10 km; $\delta^{18}$O shows a -0.008‰ m$^{-1}$
depletion which enhances the measured LGM-Holocene change in the ice core by ~1‰. This
work facilitates accurate interpretation of the SPICEcore records as temporal histories of climate
at South Pole.
**6 Data Availability**
Velocity and radar data are available at http://www.usap-dc.org/view/dataset/601100. Water
isotope, accumulation rate, and advection corrections will be posted upon acceptance.
**7 Author Contributions**
All authors contributed to the analysis and writing of the manuscript. HC, DL, MS, and MK
performed the field work. AS, TF, and ES performed water isotope analysis. TF, NH, and ES
analyzed the ice-sheet model ensemble.
**8 Competing Interests**

The authors declare no competing interests.

**9 Acknowledgements**
This work was funded through U.S. National Science Foundation grants 1443471 and 1443232 (MK, EW, HC, TJF); 1443105 and 141839 (EJS). We thank the Ice Drill Program Office for recovering the ice core; the 109th New York Air National Guard for airlift in Antarctica; Elizabeth Morton, David Clemens-Sewall, Maurice Conway, Mike Waskiewicz for their efforts in the field; Antarctic Support Contractors and the members of South Pole station who facilitated the field operations; UNAVCO for power supplies and GPS support; and the National Science Foundation Ice Core Facility for ice-core processing.

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
