# Peer review of "Advection and non-climate impacts on the South Pole Ice Core Tyler J. Fudge1, David A. Lilien1,2, Michelle Koutnik1, Howard Conway1, C. Max Stevens1,"

_Climate of the Past, 2019_

## Referee Comment (RC1) · Anonymous Referee #1 · 12 Aug 2019

The paper describes the ice-flow impact on the South Pole Ice Core. It brings essential constraints for the interpretation of this recently drilled deep ice core. While it is not the most exciting/innovative paper I have read, it uses established state of the art methods, with new observations to quantify as well as possible the impact of ice flow on the SPICE core, so that this impact would not wrongly be interpreted as a climate signal. Good care is given to the assessment of uncertainties. Overall, this is a high quality study, and it deserves fast publication.

My major comment concerns the clarity of the flow modeling method used. It is not described at all, and it deserves a bit of attention. I understand that this part of the work was done in Lilen et al. (2018), but it is central to the results presented here, and a short description of the model framework would be useful. It is difficult to understand

where results come from without this information, and as a reviewer, I am not able to judge the results without it. I am expecting a paragraph with the general framework, including inputs and outputs, major hypothesis of your flow-band model (or other type of model, I could not figure out what you used).

Line 231-238 : compare your assessment of the slope of 0.008‰ / m with the Masson-Delmotte et al. (2008) dataset.

Also, there are some copy-editing issues with the section titles : Introduction, Methods and Results don't have the same level of titles, but I presume that this will be fixed before publications. Numbering, would also be useful.

⸻⸻⸻⸻⸻⸻⸻

---

## Referee Comment (RC2) · Anonymous Referee #2 · 3 Sep 2019

General comments

This paper discusses measurements of present-day accumulation rate, water isotopes, surface velocity, and 10 m temperature in an area upstream of the SPICEcore drilled at South Pole to estimate the effect of advection on climate histories inferred from the ice core itself. The analysis is interesting and discusses many of the relevant effects one needs to take into account to separate the climate information in an ice core from effects introduced by elevation change when the ice core is drilled away from an ice divide. The analysis is however not very sophisticated as it is almost solely based on qualitative reasoning and no use is made of any ice flow modelling, which would be the more appropriate tool to quantify the advection effect. This results in a lot of hand-waving and questionable assumptions underlying the analysis.

[Figure]

More specifically I feel the paper has 3 major problems which makes the paper not publishable in its present form.

Most crucially the authors assume that accumulation rates are kept constant to their Holocene values for ice older than 10 ka, and consequently, ice velocities keep their present-day values along the presumed flowline for the last 55 ka. Even though the authors acknowledge there are good reasons to believe this is not the case (p. 14, lines 278-292), they ignore ice velocity variations citing Pollard and DeConto (2009). However, Pollard and DeConto (2009) do not have a figure in their paper showing the glacial ice flow pattern. Their supplemental material video V3 only shows details of the ice velocity during the last 8800 years. Even simple models will show that accumulation rates, and consequently ice velocities, should be roughly halved during the glacial period. This bears directly on the determination of the location of ice deposition over time, the crucial underpinning of the paper. At the very least, the authors should have presented an alternative distance traveled vs time assuming lower velocities during the glacial period and the glacial-interglacial transition. Equally the cascade of assumptions made for ice deposited beyond 70 km (constant present-day flow direction, the unconstrained straight flow line for ice older than 21 ka, the linear decrease of ice velocity for the oldest part) are so rough that the conclusion that the oldest SPICEcore ice originated ∼35 km downstream from the assumed divide position (p. 15, lines 306-307) is hard to believe.

Secondly the paper only discusses the impact of advection and ignores elevation changes from ice dynamics. Therefore absolute statements on the temperature correction required to interpret the climatic information in SPICEcore can not be made.

Thirdly, there is much overlap with Lilien et al. (2008) concerning the discussion of the measurements on which the analysis is based. Even though the focus of the current paper is different, and the time period considered is longer, this introduces unnecessary duplication of material.

Specific comments

Abstract, p. 2, lines 23-24: 'Assuming a lapse rate...'. The statement is ambiguous: the LGM-to-modern temperature change is a fixed climatological quantity, and cannot depend on the place of deposition. What is meant here is that the apparent LGM-to-modern temperature change from the core is 1.5°C lower than would have been the case if the ice was deposited locally. Also this number ignores the contribution to elevation changes from ice dynamics and should not be misinterpreted as the total temperature change.

p. 4, Figure 1: One would expect the flow directions obtained from the GPS measurements to be perpendicular to the elevation contours, but that is apparently not the case for quite a few of the arrows shown. Why is that? Errors in the drawn orientation of some of the arrows, errors in the plotted surface contours, or another genuine reason? If so, which one?

p. 4, Figure 1: why does the BedMap2 surface elevation for ITASE 07-04 deviates from the GPS measured 3090 m? Can it be a mix-up of ellipsoidal versus geoidal heights?

p. 5, line 112: the assumption is made that the there is no shearing in the upper 1750 m of the ice column. The validity of this assumption needs more discussion as it depends on the thickness of the ice along the flowline, i.e. to what fraction of the total thickness the ice was located along the flowline for a certain depth at the drill site. I would like to see the thickness along the presumed flowline together with an estimate of the depth of the deepest trajectory for the oldest SPICEcore ice at 1750 m. The authors should then at least discuss the no shearing assumption based on the vertical distribution of horizontal velocity. For isothermal ice, there is an analytical expression for the depth dependence of horizontal velocity that can be found in any textbook on ice dynamics, and this would give an estimate of the maximum possible deviation of the horizontal velocity at depth from its surface value.

p. 12, Figure 5: the red and blue lines for d18O and dD respectively, and the symbols

for individual measurements, almost overplot one another because of the respective axis scaling. For better readability, the authors could opt to show both variables separately in two adjacent plots.

p. 13, line 254: 'consistent with a dry adiabatic lapse rate'. As the authors acknowledge, the inferred surface temperature gradient is imprecise because of the duration the thermistors were left in the boreholes. Nevertheless, one would expect a higher lapse rate than dry adiabatic on the Antarctic plateau because of the strength of the surface inversion layer that increases with lower temperatures. This should be discussed. What are the implications of this rough estimate of the lapse rate for the analysis?

p. 15, line 305: 'assuming a balance velocity in an ice sheet with uniform thickness'. Please discuss how good this assumption is based on available ice thickness reconstructions/ measurements for this area. What does BedMap2 show for ice thickness along the presumed flow line? See the comment on the no shearing assumption above.

pp. 16-17: Advection impact on water isotopes. This analysis evidently ignores the effect of elevation changes over the presumed flowline for the period since 55 ka due to ice dynamics. Pollard and DeConto (2009) show the evolution of surface elevation over the glacial cycles from which a rough estimate of this effect could be made?

p. 18, lines 377-379: 'advection has enhanced the glacial-interglacial . . . by 1‰: This is only true if you assume that the present-day spatially derived elevation gradient of d18O also holds back into time. This should be discussed.

p. 18, line 379: what is the status of Steig et al. (in prep.)? Otherwise use a published reference here.

p. 18, line 381: what is 'WDC'?

p. 19, line 419: 'originated at elevations up to ∼250 m higher': this is only true for the advection part and so this inference in absolute terms cannot be made here. Elevation

changes due to ice dynamics not considered in the paper must have contributed as well to the total elevation change.

Technical

A space should be left between value and unit, e.g. 9 m instead of 9m (except for $^{\circ}$C).

---

## Author Comment (AC2) · 16 Oct 2019

Please see attached .pdf.

Please also note the supplement to this comment:
https://www.clim-past-discuss.net/cp-2019-66/cp-2019-66-AC2-supplement.pdf

---

## Author Response (AR1)

**Editor Decision: Reconsider after major revisions** (30 Oct 2019) by Barbara Stenni

Comments to the Author:

Dear Dr. Fudge,

The two referees provided a very different evaluation and in particular the Referee 2 found your paper not publishable in its present form suggesting a rejection. From your responses to referees' comments I found that your justification for not using an ice flow model for quantifying the advection impacts on isotopic and accumulation records obtained from the SPICE (South Pole) ice core could be plausible. However, there are also some other concerns regarding the possible elevation effects that are not considered in your paper and should be discussed in a revised version. My opinion is that your paper needs a deeper revision than a "normal major revision". My suggestion would be that you submit a revised manuscript answering to all the comments of both referees. This revised manuscript will be sent again to Referee 2 and also to another referee for having a further opinion.

Please, follow the instruction for authors in preparing the new manuscript. You can follow the template provided at https://www.climate-of-the-past.net/for_authors/manuscript_preparation.html. In your first version the formatting was not appropriate.

Dr. Stenni,

Please find our revised document uploaded. We have substantially revised the manuscript from what we resubmitted a few months ago. In particular, we have added an analysis of the ice-sheet ensemble of Pollard et al. (2016) for the last glacial transition. This allowed us to assess potential elevation change, as requested by referee 2, and support the assumptions of past ice flow speed. With regard to elevation change, a limited ensemble composed of the most likely parameter choices as identified in Pollard et al. (2016) shows minor elevation change (maximum change of the mean of 32 members is 26 m in the past 20 ka) and the one standard deviation uncertainty always spans changes of both signs. Therefore, we do not propose an explicit correction for past changes in elevation. With regard to past speed, we evaluate the relative change in modeled speed between 20 ka and 10ka. We find two populations in the modeled speeds, one that has little change in speed and one that has 20 ka speeds of about ~70% of the 10 ka value, similar to what one might expect given the accumulation rate change. This justifies our two choices for past speeds of keeping the speed constant and scaling with the past accumulation rate. We note that two assumptions of speed for ages older than 10.1 ka has a minor impact on the LGM-modern isotopic change, approximately 0.1‰ of a total 1.1‰ impact for $\delta^{18}O$.

We have changed the formatting of the section titles so that are all in bold of the same font size and numbered consecutively. With these changes, we believe the manuscript has addressed all of the issues raised by you and the two referees.

We have attached the original author responses to the referee comments here with updates to them in purple. We also note that the extensive revisions meant that we were not able to effectively track changes. The most significant changes are in sections 3.2 and 3.4. Figures 7 and 10 were also added.

Thank you for your consideration.
On behalf of all authors,

T.J. Fudge

The paper describes the ice-flow impact on the South Pole Ice Core. It brings essential constraints for the interpretation of this recently drilled deep ice core. While it is not the most exciting/innovative paper I have read, it uses established state of the art methods, with new observations to quantify as well as possible the impact of ice flow on the SPICE core, so that this impact would not wrongly be interpreted as a climate signal. Good care is given to the assessment of uncertainties. Overall, this is a high quality study, and it deserves fast publication.

Thank you for your review. We address your comments/suggestions below:

My major comment concerns the clarity of the flow modeling method used. It is not described at all, and it deserves a bit of attention. I understand that this part of the work was done in Lilen et al. (2018), but it is central to the results presented here, and a short description of the model framework would be useful. It is difficult to understand where results come from without this information, and as a reviewer, I am not able to judge the results without it. I am expecting a paragraph with the general framework, including inputs and outputs, major hypothesis of your flow-band model (or other type of model, I could not figure out what you used).

We have more clearly described the methodology of Lilien et al. (2018). We also describe the assumptions underlying the velocity history more clearly, including using two scenarios to better understand the sensitivity of the advection impact to the velocity assumptions.

Line 231-238 : compare your assessment of the slope of 0.008‰ / m with the Masson-Delmotte et al. (2008) dataset.

We added: Our value for $\delta^{18}O$ is in between the slope of -0.009 ‰ m$^{-1}$ from the Masson-Delmotte et al. (2008) database and the slope of -0.007 ‰ m$^{-1}$ in their multiple linear regression which includes latitude and distance from the coast.

Also, there are some copy-editing issues with the section titles : Introduction, Methods and Results don't have the same level of titles, but I presume that this will be fixed before publications. Numbering, would also be useful.

Thanks. We have numbered and formatted the section levels consistently.

We appreciate your questions and comments. In particular, the suggestion to include an advection correction where the past flow speed scales with the accumulation rate is very good. We have included it and averaged it with the constant velocity scenario to produce a preferred advection impact. Including a variable velocity scenario also helps define the uncertainty of the advection impact, which we cannot rigorously quantify but explain better with this qualitative measure. We note that the difference in advection impact at 20 ka between the two velocity assumptions is 0.1‰, roughly 10% of the 1.1‰ advection impact. Detailed responses to your questions/comments are in red below.

This paper discusses measurements of present-day accumulation rate, water isotopes, surface velocity, and 10 m temperature in an area upstream of the SPICEcore drilled at South Pole to estimate the effect of advection on climate histories inferred from the ice core itself. The analysis is interesting and discusses many of the relevant effects one needs to take into account to separate the climate information in an ice core from effects introduced by elevation change when the ice core is drilled away from an ice divide. The analysis is however not very sophisticated as it is almost solely based on qualitative reasoning and no use is made of any ice flow modelling, which would be the more appropriate tool to quantify the advection effect. This results in a lot of hand-waving and questionable assumptions underlying the analysis.

The choice to not include ice-flow modeling was made for two reasons:
1) For the first time, we were able to directly constrain a Holocene speedup using the layer thicknesses in SPICEcore and the modern accumulation pattern upstream (described in detail by Lilien et al., 2018).
2) We think our assumptions are more clearly described without an ice-flow model. For instance, we described our assumptions beyond 100 km as a linear decrease in velocity to a divide 90 km distant with an elevation as measured at the Titan Dome ITASE 07-04 core site of 3090 m. This resulted in the 54.3 ka particle originating 236 m higher.
   We have also modeled the flowband with the following assumptions:
   - Measured accumulation pattern for the first 100 km and constant accumulation beyond
   - Flux out of the flowband and flowband width both tuned to match measured velocity along the 100 km of measurements
   - No flowband width changes beyond 100 km
   - Measured bedrock topography for the first 100 km with a linear increase of 100m to the divide (to keep ice thickness approximately flat)
   - Ice temperature profile fit to measurements from IceCube array at South Pole

All of which yield the elevation of the 54.3 ka particle to have originated at 240 m higher than SPICEcore compared to 236 m higher than in the simpler description.

More specifically I feel the paper has 3 major problems which makes the paper not publishable in its present form. Most crucially the authors assume that accumulation rates are kept constant to their Holocene values for ice older than 10 ka, and consequently, ice velocities keep their present-day values along the presumed flowline for the last 55 ka. Even though the authors acknowledge there are good reasons to believe this is not the case (p. 14, lines 278-292), they ignore ice velocity variations citing Pollard and DeConto (2009). However, Pollard and DeConto (2009) do not have a figure in their paper showing the glacial ice flow pattern. Attached is a figure of the of the 10 ka speeds divided by the 20 ka speeds in Pollard and Deconto 2009 (left panel) and 2016 (right panel) model runs. We used the model output provided by the authors. We use these results to highlight that the velocity depends on many assumptions beyond the accumulation rate.

[Figure]

Figure: Speed at 10ka divided by speed at 20ka for Pollard and DeConto 2009; 2016 example model runs. Color scales are the same for both images. The green circles are the approximately flowline region of SPICEcore.

Their supplemental material video V3 only shows details of the ice velocity during the last 8800 years. Even simple models will show that accumulation rates, and consequently, ice velocities, should be roughly halved during the glacial period.

We agree that simple models show a halving of the velocity. Below is the speed at South Pole in our flowband model when we force it with a plausible accumulation scenario (see section 3.2.2) and allow the flux out of the flowband to vary at each time step to minimize changes in the surface elevation (ice thickness). However, the Pollard and Deconto runs indicate that other factors become important. This includes, but is not limited to, changes in the geometry such that there is change in the flux divergence or divide position, change in the boundary flux ultimately forced by grounding line variations, and changes in the basal boundary conditions, for instance a change from sliding to frozen bed. Therefore, we have taken your suggestion of using the velocity scaled to the accumulation rate to compare with the constant velocity scenario.

[Figure]

This bears directly on the determination of the location of ice deposition over time, the crucial underpinning of the paper. At the very least, the authors should have presented an alternative distance traveled vs time assuming lower velocities during the glacial period and the glacial-interglacial transition.
Thank you for the suggestion; it is a good one. We have included a variation of the advection correction that scales directly with an estimate of the accumulation rate history.
We also use the surface velocities from the Pollard et al. (2016) ice-sheet ensemble to support these two different choices of past speed. The limited ensemble shows two populations of speed change between 20ka and 10ka. One group shows speeds at 20 ka between 50% and 90% of the speed at 10 ka. The other group shows between no change and 10% faster speeds at 20 ka compared to 10 ka. The first group is closer to the speed that might be expected if the speed was primarily determined by the accumulation rate through a balance velocity; the second group indicates that dynamic changes are able to counteract the influence of lower accumulation rates at 20 ka. See section 3.2 and Figure 7.

Equally the cascade of assumptions made for ice deposited beyond 70 km (constant present-day flow direction, the unconstrained straight flow line for ice older than 21 ka, the linear decrease of ice velocity for the oldest part) are so rough that the conclusion that the oldest SPICEcore ice originated _35 km downstream from the assumed divide position (p. 15, lines 306-307) is hard to believe.
We have rephrased the emphasis and instead use the distance upstream from SPICEcore. We agree that the advection correction becomes increasingly uncertain for the older ages, but it also becomes increasingly small as the ice isn't flowing very fast. We have emphasized the rate of advection impact by including a third panel in Figure 8C.

Secondly the paper only discusses the impact of advection and ignores elevation changes from ice dynamics. Therefore absolute statements on the temperature correction required to interpret the climatic information in SPICEcore cannot be made.
We have chosen not to address the topic of ice-sheet elevation change and instead focus on the advection correction. We believe a tightly focused manuscript is of the greatest utility to the broad community that will interpret the SPICEcore records. We have made our reasoning more clear in the introduction with the addition of:

"The magnitude and sign of the elevation change in ice-sheet models varies depending on the specified boundary conditions, which have a large uncertainty (Pollard and Deconto, 2016; Kingslake et al., 2018). Therefore, we do not address the ice-sheet elevation change near South Pole in this paper and instead focus on the impact of ice flow on the South Pole Ice Core (SPICEcore)."

At the editor's suggestion, we have now included a detailed assessment of an ensemble of continental-scale ice-sheet models for the last glacial transition (Pollard et al., 2016) to both investigate potential changes in ice-sheet elevation and support the choices of ice velocity for ages older than 10.1 ka. This model was run at higher resolution for West Antarctica because of the uncertainty in model parameters that particularly effect this dynamic region. South Pole is within the higher resolution model domain. The output of the 625-ensemble members is shown as well as a limited ensemble of 32 members which use the parameter combinations that Pollard et al., 2016 found to be most likely. The median elevation change of the limited ensemble was small, reaching a maximum 26 m higher at 10 ka. The one standard deviation always encompassed both positive and negative changes. The full ensemble had a larger range but a similar median. Therefore, uncertainty from possible ice sheet elevation change should be considered in any interpretation of the water isotope record, but existing ice-sheet models cannot sufficiently constrain the elevation history to warrant an explicit correction. See section 3.4 and Figure 10.

Thirdly, there is much overlap with Lilien et al. (2008) concerning the discussion of the measurements on which the analysis is based. Even though the focus of the current paper is different, and the time period considered

There is significant overlap, but, as noted, the focus is distinctly different. The other referee wanted more detail on the work of Lilien et al. (2018) to support the focus of this paper which is the impact on the water isotope and accumulation histories in the ice core and we have provided it. We believe the greater explanation of the novel method employed by Lilien et al. (2018) to infer the flowline improves the manuscript and reasoning to exclude ice-flow modeling for this paper.

Specific comments

Abstract, p. 2, lines 23-24: 'Assuming a lapse rate: : :'. The statement is ambiguous: the LGM-to-modern temperature change is a fixed climatological quantity, and cannot depend on the place of deposition. What is meant here is that the apparent LGM-to-modern temperature change from the core is 1.5_C lower than would have been the case if the ice was deposited locally. Also this number ignores the contribution to elevation changes from ice dynamics and should not be misinterpreted as the total temperature change.

This sentence has been reworded. We agree that this should not be interpreted as a correction that includes any changes in the elevation of the ice sheet, which is why we do not mention the elevation history of the ice sheet in the abstract.

"Assuming a lapse rate of 10°C per km of elevation, the inference of LGM-to-modern temperature change is ~1.5°C smaller than if the flow from upstream is not considered."

p. 4, Figure 1: One would expect the flow directions obtained from the GPS measurements to be perpendicular to the elevation contours, but that is apparently not the case for quite a few of the arrows shown. Why is that? Errors in the drawn orientation of some of the arrows, errors in the plotted surface contours, or another genuine reason? If so, which one?

The primary reason that the velocities do not follow the surface contours is that the BedMap2 surface elevations are not well constrained in the "Pole Hole". They are based on a few airplane and surface traverse tracks. See Fretwell et al., 2013 (Fig 2) and the data coverage for more detail.

p. 4, Figure 1: why does the BedMap2 surface elevation for ITASE 07-04 deviates from the GPS measured 3090 m? Can it be a mix-up of ellipsoidal versus geoidal heights?
Unfortunately, no. The difference in elevation between the ITASE 07-04 core site and South Pole is 282m using the elevations of Dixon et al. (2013) Table 1. In Bedmap2, the difference is 339 m, so there is no way to reconcile the elevations with ellipsoidal versus geoidal heights. We are confused why there is this difference but wonder if there is an erroneous elevation and position in one of the flight lines included in BedMap2.

p. 5, line 112: the assumption is made that the there is no shearing in the upper 1750 m of the ice column. The validity of this assumption needs more discussion as it depends on the thickness of the ice along the flowline, i.e. to what fraction of the total thickness the ice was located along the flowline for a certain depth at the drill site. I would like to see the thickness along the presumed flowline together with an estimate of the depth of the deepest trajectory for the oldest SPICEcore ice at 1750 m. The authors should then at least discuss the no shearing assumption based on the vertical distribution of horizontal velocity. For isothermal ice, there is an analytical expression for the depth dependence of horizontal velocity that can be found in any textbook on ice dynamics, and this would give an estimate of the maximum possible deviation of the horizontal velocity at depth from its surface value.
The horizontal velocity at 1750m depth is 99% of the surface velocity. Below is a figure of the horizontal velocity calculated using two plausible temperature profiles since there is debate about whether the South Pole is at the pressure melting point. The specific temperature profile makes little difference in the fraction of surface velocity in the upper 1750 m. Using an isothermal profile would underestimate the concentration of deformation near the bed because there is nearly a 50°C change in temperature from the surface to the basal ice and the creep parameter A varies by 3 orders of magnitude. The text has been changed to more specifically state that since 99% of the deformation occurs below 1750 m, we assume a constant vertical velocity.

[Figure]

[Figure]

Trajectories for particles of 10, 20, 30, 40 and 50 ka age from flowband model forced with varying accumulation.

p. 12, Figure 5: the red and blue lines for d18O and dD respectively, and the symbols for individual measurements, almost overplot one another because of the respective axis scaling. For better readability, the authors could opt to show both variables separately in two adjacent plots.

We chose to plot them together to emphasize that they scale similarly, which is what we expect. We have attached a plot with them side by side here.

[Figure]

p. 13, line 254: 'consistent with a dry adiabatic lapse rate'. As the authors acknowledge, the inferred surface temperature gradient is imprecise because of the duration the thermistors were left in the boreholes. Nevertheless, one would expect a higher lapse rate than dry adiabatic on the Antarctic plateau because of the strength of the surface inversion layer that increases with lower temperatures. This should be discussed. What are the implications of this rough estimate of the lapse rate for the analysis?

We are not quite sure what you mean by the "strength of the surface inversion layer that increases with lower temperature." We are considering a temperature range of only ~2°C, and not the ~30°C range from the coast to the interior for which the strength of the inversion is quite important (e.g. Krinner and Genthon, 1999, Altitude dependence of the ice sheet surface climate, GRL). We believe that in our small (relative to the Antarctic ice sheet) catchment, if there is any difference in the inversion strength along the flowline, it would be driven by the regional wind pattern, which is not well constrained, rather than a temperature dependence of the inversion layer.

p. 15, line 305: 'assuming a balance velocity in an ice sheet with uniform thickness'. Please discuss how good this assumption is based on available ice thickness reconstructions/measurements for this area. What does BedMap2 show for ice thickness along the presumed flow line? See the comment on the no shearing assumption above.

There isn't much available bed data, so it's quite unclear what the ice thickness is beyond the measured flowline. Below is the bedmap2 interpolation of the bed and that inferred for the flowline and a plausible extension. For the first 40km, the bed appears very similar to the end of the flowline, which is about 200m higher in elevation than our detailed measurements.

[Figure]

Bed elevation from Bedmap2. Measured flowline in black with a plausible extenstion of the flowline.

[Figure]

pp. 16-17: Advection impact on water isotopes. This analysis evidently ignores the effect of elevation changes over the presumed flowline for the period since 55 ka due to ice dynamics. Pollard and DeConto (2009) show the evolution of surface elevation over the glacial cycles from which a rough estimate of this effect could be made?

We have intentionally avoided this topic because it is beyond the scope of this paper and is also the subject of a different paper that we are collaborating on. We have made it more clear in the manuscript that we do not address the question of ice-sheet elevation change. But to more specifically answer your question, the PD models show magnitude of ~100m elevation changes in both directions, with varying timing, depending on the choice of boundary conditions.

p. 18, lines 377-379: 'advection has enhanced the glacial-interglacial : : : by 1‰´ . This is only true if you assume that the present-day spatially derived elevation gradient of d18O also holds back into time. This should be discussed.

We have added this assumption into section 3.3.2. We note that ~70% of the LGM-modern change occurs in the Holocene, when the climate, and thus elevation-d18O gradient, is likely similar to today.

p. 18, line 379: what is the status of Steig et al. (in prep.)? Otherwise use a published reference here.

The full isotope data are not yet fully published, but relevant data are used in Kahle et al., 2018 (https://doi.org/10.1029/2018JF004764), which we now cite. The SPICEcore $\delta^{18}O$ average over the past 1000 years is -51.0‰ and the 1000 years centered at 20 ka is -57.2‰, for a difference of 6.2‰. At WAIS Divide, the difference for the same time periods is 7.1‰.

p. 18, line 381: what is 'WDC'?

WAIS Divide ice core. We apologize for excluding the full name and have now defined it on its first usage.

p. 19, line 419: 'originated at elevations up to _250 m higher': this is only true for the advection part and so this inference in absolute terms cannot be made here. Elevation changes due to ice dynamics not considered in the paper must have contributed as well to the total elevation change.

We have included the phrase "assuming a similar ice-sheet configuration in the past" at the end of the sentence.

Technical

A space should be left between value and unit, e.g. 9 m instead of 9m (except for _C).

Spaces have been added (hopefully we caught them all, but some are sneaky).